# Computational Modelling of Tunicamycin C Interaction with Potential Protein Targets: Perspectives from Inverse Docking with Molecular Dynamic Simulation

**DOI:** 10.3390/cimb47050339

**Published:** 2025-05-08

**Authors:** Vivash Naidoo, Ikechukwu Achilonu, Sheefa Mirza, Rodney Hull, Jeyalakshmi Kandhavelu, Marushka Soobben, Clement Penny

**Affiliations:** 1Department of Internal Medicine, Medicine, Wits/MRC Common Epithelial Cancer Research Centre, Faculty of Health Sciences, University of the Witwatersrand, Johannesburg 2050, South Africa; vivash.naidoo@wits.ac.za (V.N.); sheefa.mirza@wits.ac.za (S.M.); 2Department of Family Medicine and Primary Health Care, University of the Witwatersrand, Johannesburg 2050, South Africa; 3Protein Structure-Function Research Unit, School of Molecular and Cell Biology, Faculty of Science, University of the Witwatersrand, Johannesburg 2050, South Africa; ikechukwu.achilonu@wits.ac.za (I.A.); 1129969@students.wits.ac.za (M.S.); 4SAMRC Precision Oncology Research Unit (PORU), DSI/NRF SARChI Chair in Precision Oncology and Cancer Prevention (POCP), Pan African Cancer Research Institute (PACRI), University of Pretoria, Hatfield, Pretoria 0028, South Africa; rodney.hull@up.ac.za; 5Department of Oncology, Lombardi Comprehensive Cancer Center, Georgetown, University Medical Center, Washington, DC 20007, USA; biojeya@gmail.com

**Keywords:** glycosylation, Tunicamycin, thymidine kinase 1, protein kinase A, molecular dynamics, therapeutic targets, CRC (colorectal cancer), in silico techniques

## Abstract

Protein glycosylation plays a crucial role in cancer biology, influencing essential cellular processes such as cell signalling, immune recognition, and tumour metastasis. Therefore, this study highlights the therapeutic potential of targeting glycosylation in cancer treatment, as modulating these modifications could disrupt the fundamental mechanisms driving cancer progression and improve therapeutic outcomes. Recently, Tunicamycin C, a well-known glycosylation inhibitor, has shown promise in breast cancer treatment but remains unexplored in colorectal cancer (CRC). Thus, in this study, we aimed to understand the potential action of Tunicamycin C in CRC using in silico studies to identify possible drug targets for Tunicamycin C. First, we identified two target proteins using the HTDocking algorithm followed by GO and KEGG pathway searches: thymidine kinase 1 (TK1) and cAMP-dependent protein kinase catalytic subunit alpha (PKAc). Following this, molecular dynamics modelling revealed that Tunicamycin C binding induced a conformational perturbation in the 3D structures of TK1 and PKAc, inhibiting their activities. This interaction led to a stable design, promoting optimal binding of Tunicamycin C in the hydrophobic pockets of TK1 and PKAc. Serial validation studies highlighted the role of active site residues in binding stabilisation. Tunicamycin C exhibited high binding affinity with TK1 and PKAc. This study provides a way to explore and repurpose novel inhibitors of TK1 and PKAc and identify new therapeutic targets, which may block glycosylation in cancer treatment.

## 1. Introduction

Despite global population-based screening efforts, colorectal cancer (CRC) universally remains a public health concern, ranking second in cancer-related mortality in developed countries [1,2]. Although the exact causes and risk factors for CRC are still not fully understood, research has focused on environmental, genetic, epigenetic, and post-translational modifications (PTMs) of proteins as potential risk factors [3]. Despite advancements in treatment regimens, the prognosis of metastatic CRC is poor, with a 5-year survival rate of less than 15% [4,5]. In recent years, there has been a shift towards finding specific molecular targets for cancer treatment, known as targeted therapy. However, this approach has limitations, as it is only effective in a small percentage of patients, with some patients developing resistance over time [6]. Therefore, further research is required, to overcome these challenges and improve the success rate of targeted therapy.

Considering this, identifying reliable molecular targets and developing novel targeted cancer therapies are crucial. Understanding the molecular mechanisms in cancer cells and their complex cellular interactions forms the basis for rational drug design. Among the critical molecular changes in cancer, altered glycosylation is a prominent and well-established hallmark. It represents the most prevalent form of post-translational modification (PTM) of proteins. It plays a significant role in cancer progression and influences cellular processes such as cell signalling, adhesion, and immune evasion [7,8,9,10,11]. Abnormal glycosylation patterns have been linked to various oncogenic processes, such as cell growth and adhesion, increased motility, apoptosis inhibition, and metastasis [1,12,13,14,15,16]. Considering this, altered glycosylation holds promise as a potential target in drug discovery.

Tunicamycin C is a natural product from *Streptomyces lysosuperificus*. It is a nucleoside antibiotic with antibacterial, antiviral, and anticancer activities. Tunicamycin C significantly inhibits the transfer of active sugars to dolichol phosphate, an essential step in the *N*-glycosylation of proteins in the endoplasmic reticulum [17,18,19,20]. Several studies have explored the potential of Tunicamycin C as a therapeutic anticancer drug by inhibiting the synthesis of N-linked oligosaccharides in different cell types. Tunicamycin C decreases the expression of membrane receptors associated with carcinogenicity and enhances sensitivity to therapy in lung cancer cells by modulating the AKT/NF-κB signalling pathway [21].

Additionally, Tunicamycin C inhibits the glycosylation of plasma membrane receptors, impairing their transport to the cell surface. In the HCT-116 CRC cell line, low doses of Tunicamycin C induced functional E-cadherin-mediated cell–cell adhesion, inhibiting cell proliferation and the development of a differentiated-like phenotype [22]. Furthermore, Tunicamycin C has been demonstrated to arrest under glycosylated FLT3-ITD in the endoplasmic reticulum and promote STAT5 activation in FLT3-ITD mutant cells in acute myeloid leukaemia [23]. Also, Banerjee et al. suggested Tunicamycin C as an excellent glycotherapy in breast cancer [24]. These findings altogether highlight its diverse therapeutic applications in various cancer types. Nevertheless, the therapeutic potential of this strategy remains unclear and warrants further investigation, particularly in CRC. As the mechanism of action and targets are still not fully explored, in this study, using an in silico investigation, we sought to identify targets for Tunicamycin C that may play a role in glycosylation in CRC.

Initially, two target proteins, TK1 and the catalytic subunit of cAMP-dependent protein kinase (PKAc), were identified based on their role in different cancer pathways through manual curation, although their direct involvement in glycosylation still requires elucidation. To further investigate these targets, we employed a computational approach to unravel the mechanisms and dynamics surrounding this selective binding interaction of Tunicamycin C with these two targets. This has enabled us to understand the structural events that promoted the binding inhibition of PKAc and TK1, potentially ameliorating cancer and aiding in the structural design of more effective drugs for treating CRC.

## 2. Materials and Methods

Figure 1 shows the workflow that was incorporated in this study. This study aims to explore Tunicamycin C as a potential therapeutic agent targeting colorectal cancer (CRC) and glycosylation pathways. The study begins with identifying potential molecular targets associated with CRC and glycosylation using two methods, SwissADME Prediction and CBLigand High-Throughput Docking. Pathway analysis was carried out using KEGG pathway analysis. Afterwards, pathways involved in CRC and glycosylation were identified, and suitable targets were selected. Molecular dynamic simulation modelling and docking studies assessed the interactions between Tunicamycin C and the selected target proteins. These studies were complemented by thermodynamic binding free energy calculations to evaluate the strength and stability of Tunicamycin C binding to the targets. A literature review was carried out throughout this process to provide context and validate the findings. This multi-step approach integrates cheminformatics, molecular modelling, and computational biology to investigate the therapeutic potential of Tunicamycin C in CRC treatment, focusing on its molecular interactions and involvement in glycosylation-related pathways.

### 2.1. Identification of Target Proteins

The Chemical Entities of Biological Interest/ChEBI database (https://www.ebi.ac.uk/chebi/, accessed on 1 April 2025) is a website that provides molecular information on small, potentially active biological chemicals. The simplified molecular input line entry system (SMILES) file of Tunicamycin C was also obtained from the ChEBI database and was submitted to SwissTarget to detect additional protein interactors. These analyses identified several proteins that may interact with Tunicamycin C; these protein targets were screened for hits that matched a crucial role in regulating glycosylation in CRC. To verify protein–ligand interactions, the proteins were further screened using CBLigand High-Throughput Docking. The High-Throughput Docking program (https://www.cbligand.org/HTDocking/search_struct.php, accessed on 16 May 2022) was utilised to identify proteins that may interact with Tunicamycin C and determine the strength of the interactions. The downloaded .sdf file of Tunicamycin C was uploaded for analysis to the HT docking site. The targets with positive hits were selected for further studies and analysis. The docking score is used to approximate the binding affinity between the two molecules, with a higher binding score reflecting higher affinity [25].

The KEGG pathway analysis (https://www.genome.jp/kegg/pathway.html, accessed on 1 April 2025) database was used to identify pathways involving these proteins and to assign Gene Ontology terms. During the KEGG analyses, *p* < 0.01 was deemed statistically significant. A literature review was then utilised to identify the proteins connected with cancer, specifically CRC. This list was carefully screened with support from the biomedical literature to identify such targets. To validate the choice, putative protein targets were further verified for their available high-quality 3D structures at the Protein Data Bank (PDB). These ligand-based techniques identified protein kinase A-catalytic subunit alpha (PKAc) and thymidine kinase 1 (TK1) as potential target proteins for Tunicamycin C (Appendix A).

### 2.2. Computational Modelling Studies

#### 2.2.1. Computer Hardware Used for Computational Modelling Studies

All molecular modelling studies were conducted using two high-performance computing units. The first system, running Windows OS, featured the Maestro algorithm installed on a PC powered by an AMD RYZEN Threadripper 1950X processor, an Asus Rog Strix X399-E motherboard, and 64 GB of DDR4 RAM (3200 MHz). This system included a 4 TB SSD with read/write speeds of 560 MB/s and 520 MB/s, respectively, and a GeForce RTX 2080 Ti graphics card (11 GB GDDR6). The second system, running Ubuntu OS, had the Desmond molecular dynamics simulation algorithm installed. It used an AMD Threadripper 3990X processor with 64 cores, an MSI TRX40 PRO motherboard, 64 GB of 3200 MHz RAM, a GeForce RTX 2070 graphics card (8 GB GDDR6), a 1 TB M.2 SSD (3.5 GB/s speed), and a 4 TB HDD for additional storage.

#### 2.2.2. Preparation of TK1, PKAc, and Tunicamycin C for Induced Fit Docking and MD Simulation

The cytosolic thymidine kinase’s (TK1) three-dimensional structure coordinates were extracted from the open-access digital resource, the RCSB Protein Data Bank (PCSB PDB), using a PDB ID: 1XBT. Similarly, the three-dimensional coordinates of PKAc were extracted from the same resource using a PDB ID: 3AMA. Both protein target coordinates obtained were subsequently submitted to the Protein Preparation Wizard module implemented in the Maestro v13.0 molecular modelling algorithm. Subsequently, hydrogen atoms were added, zero-bond orders were placed on metals, which will prevent any covalent interaction between the molecules and any metal ion placed in the periodic boundary, and disulfide bonds were created. Optimisation of the hydrogen-bonding network was achieved by sampling the water orientation with the PROPKA algorithm at pH 7.0. Model modification was carried out by correcting bond order and removing any bound ligand in the structure, in addition to water molecules whose hydrogen atoms are within the range of 5 Ả.

Refinement of the model, until an average RMSD value of 0.3 Å was reached, was performed using the OPL_2005 force field. This was followed by assessing the side chain’s stereochemistry to ensure that an absence of any significant perturbation was induced during structure preparation. With the use of a database of chemical molecules’ activities against their biological assays, PubChem was used to extract structure data files (SDF) belonging to Tunicamycin C (CID:1622051). These were then subsequently submitted (for ligand preparation) to the LigPrep and Epik modules implemented in Maestro v13.0. The use of these tools was to ensure energy minimisation with the use of the OPLS_2005 force field, bringing about ligand desalting, and possibly generate each ligand’s tautomeric state at pH 7.0 ± 2. Associated with this was the accurate prediction of the pKa of these states at the established pH with the use of the Epik module of the algorithm. Furthermore, some specific chiral centres were either varied or retained during ligand preparation to return low-energy states of chemically sensible structures. The generated molecule of each ligand was saved separately as Maestro (.mae) files for subsequent use for induced fit ligand docking.

#### 2.2.3. Docking Tunicamycin C into TK1 and PKAc Using Induced Fit Ligand Docking

The Schrödinger Maestro v13.0 algorithm’s implementable induced fit docking was used to predict the binding of Tunicamycin C to TK1 and PKAc models. This is imperative since the ligand’s binding site in a protein primarily depends upon conformational changes that the ligand induces in the protein structure upon binding. Since the tetrameric TK1 structure is known for each subunit containing an α/β domain [26], all four possible binding sites were then specified as binding sites for Tunicamycin C. However, due to the heterodimeric structure-forming ability of PKAc, binding neither as a homodimer nor a monomer [27], two possible binding sites were specified as binding sites for PKAc. The induced fit ligand process with ring conformational sampling was carried out using the OPLS4 force field with both an implicit solvent model and a 2.5 kcal/mol energy barrier with a non-polar conformational penalty on amide bonds. The receptor and ligand scale were set at 0.5, with 20 maximum allowable poses per ligand. A post-refinement procedure was carried out on residues within 5.0 Ả of the docked ligand using the prime refinement module algorithm (implemented in Maestro v13.0) to further rank the refined protein–ligand complexes. Subsequently, to conduct a final round of Glide docking and scoring, the default Glide XP setting was used. This was achieved by resubmitting the receptor structures within 30.0 kcal/mol of the minimum energy structure with the re-docking of each ligand into every single refined low-energy receptor structure in the accompanying second docking step.

#### 2.2.4. Molecular Dynamics Simulation Studies

Molecular dynamics (MD) simulations were conducted using the GPU-enabled Desmond simulation engine in Maestro v13.0. The top scoring poses of Tunicamycin C (TunC) in complex with TK1 and PKAc, along with the apo forms of both proteins, were saved as PDB files and submitted to a Linux (Ubuntu) desktop server for simulation. Before the simulations, the System Builder module in Desmond was used to prepare the four systems (TunC:TK1, TunC:PKAc apo-TK1, and apo-PKAc). This step involved solvating the system with the TIP3P explicit solvent model and using the OPLS4 force field. Each system was placed in an orthorhombic box with a 10 Å distance from the box edge to the outermost atom, and the box angles were set to *α* = *β* = *γ* = 90°. The system was minimised, and counter ions (positioned at least 20 Å from the ligand) were added to neutralise the charge. A physiological condition of 0.15 M NaCl was added to the solvent box.

After solvation and ionisation, the system advanced to the molecular dynamics production phase, divided into eight stages. Stages 1–7 focused on equilibration with short simulation steps, while stage 8 was the final 50 ns long simulation. In stage 1, the system type and parameters were detected. Stage 2 involved a 100 ps simulation under NVT conditions at 10 K with restraints on heavy solute atoms using Brownian Dynamics. Stage 3 followed with a 12 ps simulation under NVT conditions at 10 K, also with restraints on heavy atoms. Stages 4, 6, and 7 used short simulation steps (12, 12, and 24 ps, respectively) under NPT conditions at 10 K, with heavy atom restraints in stages 4 and 6 but none in stage 7. Stage 8 was the final MD production phase, conducted at a constant temperature of 300 K.

#### 2.2.5. Post-Dynamic Analysis

The molecular dynamics simulation trajectories were analysed post-simulation using Schrodinger Maestro v13.0. Specifically, the quality of the simulation was evaluated by examining the root mean square deviation (RMSD) of the alpha carbon atoms and the ligand–receptor complex. Additionally, root mean square fluctuations (RMSFs) of the residues, secondary structure elements, and protein–ligand interactions were analysed using the Simulation Interaction Diagram algorithm in Maestro v13.0. The radius of gyration (Rg) and atomic distance calculations were determined using the Simulation Events Analysis algorithm, also within Maestro v13.0.

To analyse the molecular dynamics data of PKAc and TK1 proteins in their apo form and when treated with Tunicamycin C (**1**), Principal Component Analysis (PCA) was conducted on the C_α_ RMSD, C_α_ RMSF, and RoG parameters. R Studio (Posit Software, PBC Version 2023.09.1 + 494) along with the ggfortify package [28,29] was used to highlight the apo form and Tunicamycin C-treated form. This analysis allowed for the comparison of structural changes induced by Tunicamycin C, providing insights into the dynamic behaviour of each protein.

#### 2.2.6. Implicit Physiological Condition MM/PBSA Binding Free Energy Calculation

Topology and parameter files for the complexes were generated using the LEAP module, and counter ions were added at a constant pH to neutralise the system. The systems were then solvated in a 10 Å TIP3P water box (Figure 1). Initially, partial minimisation was performed for 2500 steps with a restraint potential of 500 kcal/mol Å, followed by full minimisation for 5000 steps without any energy restraints. The systems were gradually heated from 0 to 300 K using a Langevin thermostat and a harmonic restraint of 5 kcal/mol Å^2^ in an NPT canonical ensemble for 50 ps. Equilibration was carried out at 300 K for 1000 ps without energy restraints, while atmospheric pressure was maintained at 1 bar using the Berendsen barostat. A production MD simulation was then run for 25 ns, with the SHAKE algorithm restraining all atomic hydrogen bonds. The resulting trajectories were analysed using the integrated CPPTRAJ and PTRAJ modules, with coordinates saved at 1 ps intervals. Data were plotted and analysed using Origin software 2023b (10.05). Additionally, 3D structural visualisation and analysis were performed using the UCSF Chimera graphical interface.

The Molecular Mechanics/Poisson–Boltzmann Surface Area (MM/PBSA) method was employed to estimate small therapeutic molecules’ binding interaction free energy with biological macromolecules. This study computed the binding interaction free energy from the last 25 ns trajectory. Mathematically, binding free energy is represented by the following equation:ΔG_bind_ = G_complex_ − G_receptor_ − G_ligand_(1)E_gas_ = E_int_ + E_vdw_ + E_ele_(2)G_sol_ = G_PB_ + G_SA_(3)G_SA_ = γSASA(4)

The equation above shows that E_gas_ represents the gas-phase energy, while E_int_ denotes the internal energy. The Coulombic and van der Waals energies are represented by E_ele_ and E_vdw_, respectively. Additionally, G_sol_ refers to the free energy of solvation, with the polar solvation contribution denoted by G_PB_. In contrast, G_SA_ accounts for the non-polar solvation contribution, which is estimated based on the solvent-accessible surface area (SASA) using a water probe with a radius of 1.4 Å and a surface tension constant (*γ*) of 0.0072 kcal/(mol·Å^2^). Furthermore, a per-residue decomposition analysis was performed to determine the energy contributions of individual binding site residues to ligand affinity and stabilization.

## 3. Results

Figure 2a shows the structure of Tunicamycin C (**1**). The compound is a nucleoside antibiotic characterised by a structure that includes a tunicamine moiety (an aminoglycoside sugar) linked to uracil and a fatty acid chain. The core structure consists of a glycosylated uracil unit where the sugar is an *N*-acetylglucosamine analogue. The key components of the structure (the uracil, the modified sugar, and the fatty acid chain) contribute to its biological activity, especially its ability to inhibit *N*-acetylglucosamine transferases involved in the glycosylation processes. This inhibition disrupts protein glycosylation, crucial for the antibiotic’s potential therapeutic effects against cancer [30].

Figure 2b,c show the chemical structures of two compounds, Triflourothymidine (**2**) and Bisindolylmaleimide I (GF109203X) (**3**), which act as inhibitors for TK1 and PKAc (potential Tunicamycin C targets selected in this study). Trifluorothymidine (TFT) (**2**), or Trifluridine, is a fluorinated pyrimidine nucleoside analogue structurally similar to thymidine. Its structure comprises a thymine base in which the methyl group at the 5-position is replaced by a trifluoromethyl group (CF3). Like thymidine, the base is attached to a deoxyribose sugar through a beta-*N*-glycosidic bond. This trifluoromethyl substitution enhances the molecule’s lipophilicity and increases its metabolic stability, making it more effective as an antiviral and anticancer agent. The trifluoromethyl group disrupts regular base pairing during DNA synthesis, leading to cytotoxic effects, especially in rapidly dividing cells. Bisindolylmaleimide I (GF109203X) (**3**) serves as a protein kinase A inhibitor, targeting the catalytic subunit (PKC) [31]. Its structure is characterised by a central maleimide core attached to two indole rings (Figure 2c), conferring the ability to bind to the ATP-binding site of PKA, thus inhibiting its phosphorylation activity [32]. It is worth noting that PKC is among the AGC kinases, named after PKA, PKC, and PKG families, and comprises over 60 serine/threonine kinases classified into 14 subfamilies. Regulated by lipids and cyclic AMP, they are critical for cellular functions. Their conserved structure and hydrophobic catalytic motif link them to diseases like cancer and diabetes upon dysregulation [33].

### 3.1. Screening Results

#### Screening Potential Tunicamycin C Protein Targets

The SMILES (simplified molecular input line entry system) of Tunicamycin C, Trifluorothymidine, and Bisindolylmaleimide I was utilised in SwissTarget protein screening. A total of 93 protein targets were obtained from SwissTarget Prediction. Table 1 illustrates some of the protein targets for Tunicamycin C and their respective known actives in 3D and 2D, as the Swiss Protein Predict tool predicted. Known actives in 3D and 2D provide insight into the potential interactions and binding affinities between Tunicamycin C and the protein targets. A higher number of known actives suggest that the compound may have a stronger affinity for the specific protein target and, thus, a higher likelihood of modulating its function.

The spatial data files (.sdf) for Tunicamycin C, Trifluorothymidine, and Bisindolylmaleimide I were obtained from the PubChem database (https://pubchem.ncbi.nlm.nih.gov/, accessed on 1 January 2020). This website provides molecular information on small, potentially active biological chemicals. Since the probabilities of all the top hits were zero, Tunicamycin C, Trifluorothymidine, and Bisindolylmaleimide I were subjected to LigPrep ligand energy minimisation in an implicit solvent, which generated several possible conformers at pH 7.0 ± 0.2 using the OPLS4 force field implemented in Maestro v13. During implicit solvent minimization at pH 7.2, the ligands achieved a lower-energy, solvent-stabilised conformation that reflects its interactions with the surrounding solvent environment. Charges in the molecule stabilised due to the dielectric properties of the implicit solvent model, ensuring electrostatic balance. Additionally, unfavourable steric clashes and strained bond geometries were relieved, reducing internal strain. This process enhanced the environment-specific stability of Tunicamycin C, preparing it for subsequent docking analyses.

The resulting energy-minimised ligands were presented to the High-Thoughput Docking algorithm to elucidate the target most likely to interact with the ligands. The ligands were uploaded to the High-Throughput Docking website (https://www.cbligand.org/HTDocking/searchstruct.php, accessed on 16 May 2022) to assess likely protein interactions with Tunicamycin C. A high docking score suggested a strong ligand and protein interaction (Table 1). Combining the Swiss Protein Predict, HTDocking, and KEGG analysis (Table 2), two putative targets were selected: PKAc and TK1 (Table 1). PKAc was implicated in several cancer pathways, and an HTDocking score of 6.8 was observed. TK1 was chosen because it is a novel therapeutic target currently being investigated for CRC [34] (Table 1). An HTDocking score of 7.4 was observed (Table 1). The controls were also run through both the target prediction models, and Trifluorothymidine was the best and first hit on SwissTarget Prediction, which was validated in the CBLigand high-throughput docking program, which scored an HTDocking score of 6.5.

KEGG pathway analysis was carried out on 93 protein targets to elucidate the molecular mechanisms of Tunicamycin C inhibiting pathways involved in colorectal cancer and glycosylation (Table 2) using the STRING database [35]. In total 108 enriched KEGG pathways met the FDR screening threshold ≤ 0.05 (Appendix A). The pathways involved in CRC and glycosylation were selected and presented in Table 2. The Ras signalling pathway is the most significantly enriched pathway involved in CRC, with 11 genes implicated and a highly significant false discovery rate (FDR) of 7.94 × 10^−7^. Key genes involved include FLT3, FGF2, PRKACA, and PKAc. The proteoglycans in the cancer pathway also show significant involvement, with 10 genes and an FDR of 1.75 × 10^−6^, encompassing genes such as MMP2, FGF2, and PKAc, which are associated with extracellular matrix remodelling and tumour microenvironment regulation.

Additionally, the general pathways in cancer involve 13 genes (FDR of 2.49 × 10^−5^), including oncogenes and signalling molecules like PKAc, AKT3, and BCL2, which play essential roles in cellular growth and survival mechanisms. Pathways linked to drug resistance, such as EGFR tyrosine kinase inhibitor resistance, include six genes with an FDR of 5.42 × 10^−5^, suggesting potential therapeutic targets. The PI3K-Akt and MAPK signalling pathways, both central to cell proliferation and survival, showed involvement with six and seven targets, respectively, with moderate FDRs, pointing to their interconnected roles in oncogenic signalling. Additional pathways, such as central carbon metabolism in cancer, VEGF signalling, Wnt signalling, and JAK-STAT signalling, were enriched. In addition, KEGG pathways related to glycosylation processes were examined, with galactose metabolism and starch and sucrose metabolism showing significant gene enrichment (FDRs of 7.94 × 10^−7^ and 4.60 × 10^−8^, respectively), involving genes like HK2, GAA, and B4GALT1. The O-glycan biosynthesis pathway, with three associated genes and a less significant FDR, highlights specific aspects of post-translational modifications relevant to cellular signalling and cancer.

### 3.2. Molecular Dynamics Simulation and Post-Dynamic Analysis

The 3D conformational perturbation of TK1 and PKAc was computed using the Cα- root mean square deviation (Cα-RMSD) and Cα root mean square fluctuation (Cα-RMSF). The Cα-RMSD is a predictive analysis used to understand the structural stability of proteins by evaluating their backbone atoms concerning their starting structures during the MD simulation. A high Cα-RMSD value indicates elevation in structural variation, which correlates with instability, whereas low Cα-RMSD corresponds to a structurally stable protein [36,37]. Figure 3 (top panel) depicts the apo TK1 and Tunicamycin C bound-TK1 trajectory patterns of their Cα-RMSD. Both the apo TK1 and the Tunicamycin C-bound systems closely follow each other, an indication that Tunicamycin C did not cause noticeable structural instability of TK1. Similarly, the Cα-RMSF trajectory in Figure 3 (top right panel) shows that Tunicamycin C caused little or no fluctuation in the TK1’s residue fluctuation patterns. On the other hand, the Tunicamycin C-bound PKAc causes significant structural instability of PKAc. Here, Figure 3 (bottom left panel) demonstrates that the Cα-RMSD trajectory pattern of Tunicamycin C-bound PKAc was higher than that of the apo PKAc. This indicates that Tunicamycin C is binding to PKAc and possibly inhibiting it. The Cα-RMSF of Tunicamycin C-bound PKAc also shows a noticeably higher fluctuation in the first 24 residues compared to apo PKAc. Previous studies have indicated that RMSF relates to functional activity, while rigidity indicates inactivity [36,37].

A 3D image showing how Tunicamycin C fits properly in the binding pockets of both TK1 and PKAc is presented in Figure 4. Both the cartoon and surface representations provide a clear and smart view of the ligand conformational alignment and orientation at both pockets. Figure 4 (top panel) shows the cartoon and surface representations, respectively, of Tunicamycin C strongly binding at the interface of TK1. In comparison, Tunicamycin C binds at a single pocket of PKAc, as depicted in the respective cartoon and surface representations in Figure 4 (bottom panel).

#### 3.2.1. Comparative Evaluation of Structural Drift from Their Centre of Mass

The radius of gyration (Rg) is an insightful way of deducing the mobility of systems, which is the structural displacement of systems from the individual centre of mass (Figure 4). It measures the degree of structural compactness during the MD simulation. Observations made from previous investigations have revealed that lower Rg values are an indication of a structurally compact system which translates to system stability, while the reverse is the case when Rg is higher [36,37]. Figure 4 shows that Tunicamycin C causes a structural change in both TK1 and PKAc as there were no mobility comparisons of the apo and bound Tunicamycin C TK1 and PKAc systems.

#### 3.2.2. Estimation of Relative Tunicamycin C Binding Dynamics for TK1 and PKAc

From the plot of the ligand RMSD (Figure 5) for the binding of Tunicamycin C to TK1 and PKAc, the plots of the ligand RMSD of both systems maintained a seemly flat trajectory after attaining equilibrium around 20 ns. At about 140 ns of the MD simulation, although a fluctuation in the two systems occurred, these nevertheless stabilised down to the end of the MD simulation. Other ligand parameters were studied with the plots of MSA and SASA.

Figure 5 shows that the orientation of Tunicamycin C in complex with PKAc maintained a seemingly steady configuration during the 250 ns MD simulation, judging from the superposed images; this buttresses the explanation given to the ligand RMSD plot. Figure 5 shows that Tunicamycin C, bound to TK1, established an excellent interaction that produced the observed consistent conformational orientation of Tunicamycin throughout the 250 ns MD simulation. Molecular surface area (MSA) Figure 5 (MolSA) is an analytical method that describes the degree to which a compound establishes an interaction with its molecular environment. Investigations have revealed that the higher the MSA, the more the ligand interacts with its surrounding amino acid residues.

Hydrophobicity should promote Tunicamycin C binding since it has a large carbon hydrophobic region, which should interact better in a hydrophobic environment. Moreover, hydrophobicity increases protein–ligand binding proficiency. A relative overall alteration in the hydrophobicity of Tunicamycin C when bound to TK1 and PKAc was estimated (Figure 6). This was performed by calculating SASA, a parameter that describes the transition and mobility of moieties across the solvent surfaces and the hydrophobic regions. An increase in SASA might imply that the binding pocket is expanding or becoming more accessible, which could affect the stability of the ligand–protein interaction. This might suggest partial or complete dissociation of the ligand or indicate increased flexibility in the binding site. On the other hand, a decrease in the SASA of a ligand generally indicates that it has become more tightly associated with its receptor or adopted a conformation that reduces solvent exposure, suggesting enhanced stability in the binding site or stronger binding affinity [38].

A time-based structural presentation of ligand conformation and orientation during the MD simulation shows that Tunicamycin C was not tightly bound to TK1 but remained tightly bound to PKAc throughout the 250 ns simulation time. This is evident in the flexible conformational orientation of the binding of Tunicamycin C to TK1 (Figure 7A) and the tightly bound conformational orientation of Tunicamycin C to PKAc (Figure 7B).

### 3.3. System-Based Visualisation of Ligand Interaction

The 2D interaction diagram of the complex of Tunicamycin C with TK1 and PKAc is presented in Figure 8; this demonstrates the comparative sampling of interaction patterns of Tunicamycin C with TK1 and PKAc in the docking conformation (Figure 8A,C) and the conformation during the MD run (Figure 8B,D). Also, it indicates the role of bulk water in promoting an adequate interaction of Tunicamycin with TK1 and PKAc. An additional analytical step was implemented to enhance our understanding of the primary active site residues crucial for the stable binding of Tunicamycin within the pockets of TK1 and PKAc (Figure 9). This process involved predicting these interactions by examining the average structural results of the complexes throughout the 250 ns MD simulation. This analysis relied heavily on assessing the mean PDB structure, demonstrated in Figure 9.

The detailed interactions between Tunicamycin atoms with TK1 (A) and PKAc (B) residues are depicted in Figure 9, where interactions present for more than 30% of the simulation time (0.00 to 250 ns) are highlighted. 

### 3.4. Quantitative Estimation of Implicit Interaction Contributors

Estimation of the most predominant types of binding interaction stabilising Tunicamycin was evaluated with stacked bar charts (Figure 10). This figure presents two stacked bar charts illustrating the interactions stabilizing Tunicamycin C (TunC) with (i) TK1 and (ii) PKAc. Each bar represents the fraction of the simulation time during which a specific residue interacts with TunC, categorised by interaction type. Green bars indicate hydrogen bonds, which are critical for binding specificity and stability, with significant contributions observed in both TK1 and PKAc. Blue bars represent water bridges, reflecting water-mediated interactions that enhance flexibility and are notably prevalent in PKAc. Red bars depict ionic interactions, though these are relatively sparse, suggesting fewer direct electrostatic interactions. Grey bars highlight hydrophobic interactions, underscoring non-polar stabilisation, with some residues showing significant contributions to the binding. Overall, the interaction patterns reveal diverse mechanisms stabilising the TunC complexes with TK1 and PKAc, reflecting their distinct binding environments and functional roles.

Different quantitative approaches were explored to understand the conformational events that lead to the obtained structural dynamics (Figure 10). We took a step further to disclose and understand the nature and types of forces, which provided the interactions that produced the binding stabilisation of Tunicamycin on TK1 and PKAc. This figure (Figure 11) shows the per-residue energy decomposition for TK1 (panel A) and PKAc (panel B), highlighting the contributions of specific residues to the stabilisation of Tunicamycin. Energy contributions are divided into total energy (purple), electrostatic energy (green), and van der Waals (vdW) energy (orange). Several residues exhibit significant contributions in both TK1 and PKAc, with notable differences in the relative importance of electrostatic and vdW interactions for each protein. For TK1 (panel A), residues such as Arg42, Arg39, and Gln220 contribute strongly through electrostatic interactions, while others like Val147 and Phe157 have significant vdW contributions. In PKAc (panel B), residues such as Tyr330 and ASP166 show prominent electrostatic contributions, while hydrophobic residues like Phe129 and Leu173 stabilise through vdW interactions. When related to Figure 11, it becomes clear that residues contributing strong energy in the first figure are also involved in specific stabilising interactions. For example, Arg42 and Asp166 show prominent ionic interactions, while hydrophobic residues contribute both vdW energy and hydrophobic contacts. This alignment reinforces the critical role of these residues in stabilising Tunicamycin binding. Figure 1 and Figure 11 provide a comprehensive understanding of the binding dynamics and energy contributions for both TK1 and PKAc.

### 3.5. Quantification Evaluation of Total Binding Free Energy of TK1 and PKAc

The binding free energy shows the MD simulation timeline’s silent interaction and dynamic trends. MM/GBSA was employed here to gain insight into the binding affinity of Tunicamycin C to TK1 and PKAc. Table 2 summarises the energy components of molecular interactions between the protein complexes TK1_TunC and PKAc_TunC with the ligand Tunicamycin C, highlighting their contributions to the binding free energy (Δ*G*_bind_) in kcal/mol. TK1 exhibits stronger van der Waals interactions (Δ*E*_vdw_ = −69.13 kcal/mol) compared to PKAc (Δ*E*_vdw_ = −37.05 kcal/mol), while PKAc shows stronger electrostatic interactions (Δ*E*_ele_ = −110.32 kcal/mol) than TK1 (Δ*E*_ele_ = −70.08 kcal/mol), albeit with higher variability. Both complexes demonstrate favourable gas-phase interactions (Δ*G*_gas_), with PKAc being slightly more favourable (−147.37 kcal/mol) than TK1 (−139.20 kcal/mol). However, solvation-free energy (Δ*G*_sol_) is less favourable for PKAc (106.78 kcal/mol) than for TK1 (78.99 kcal/mol), diminishing the overall binding affinity of PKAc. Consequently, TK1 achieves stronger binding (Δ*G*_bind_ = −60.21 kcal/mol) than PKAc (Δ*G*_bind_ = −40.59 kcal/mol), suggesting that TK1 has a more favourable interaction with Tunicamycin C (Table 3).

### 3.6. Molecular Dynamics Data Analysis of PKAc and TK1

Principal Component Analysis of PKAc protein C_α_ RMSD, C_α_ RMSF, and RoG reveals significant structural and dynamic changes in response to the Tunicamycin association. The PCA plot for C_α_ RMSD (Figure 12A) shows clear separation between the apo (black) and the Tunicamycin-induced form (red), indicating significant conformational changes, with PC1 accounting for 74.49% of the variance and PC2 accounting for 25.51%. The C_α_ RMSF PCA plot (Figure 12B) indicates some overlap, but with distinct shifts between the apo and bound forms, with PC1 accounting for 89.96% of the variance and PC2 explaining 10.04%, thus suggesting increased flexibility in certain regions of the Tunicamycin-bound forms. The RoG PCA plot shown in Figure 12C accentuates the distinct clustering for the apo and Tunicamycin-bound forms, with PC1 accounting for 98.66% of the variance and PC2 accounting for 1.34% of the variance. This indicates variations in protein compactness in response to the presence of Tunicamycin within the protein complexes.

PCA of the TK1 protein shows significant structural and dynamic changes resulting from Tunicamycin C treatment. The C_α_ RMSD PCA plot (Figure 13A) shows a distinct separation between the apo (black) and the Tunicamycin-treated (red) protein, indicating notable conformational changes, with PC1 accounting for 83.86% of the variance and PC2 accounting for 16.14%. The C_α_ RMSF PCA plot (Figure 13B) shows some overlap but also distinct shifts, with PC1 explaining 90.75% of the variance and PC2 explaining 9.25%, suggesting increased flexibility in certain regions of the Tunicamycin-treated forms. The RoG PCA plot (Figure 13C) highlights distinct clustering for the apo and Tunicamycin-treated forms, with PC1 accounting for 80.70% of the variance and PC2 accounting for 19.30%, indicating changes in protein compactness upon treatment. Overall, the PCA results demonstrate that Tunicamycin C significantly affects the structural integrity and dynamics of TK1, leading to notable conformational changes, localised increases in flexibility, and variations in compactness, potentially impacting the biological functioning of the protein.

## 4. Discussion

In this study, we focused on PKAc, a member of the AGC kinase family pivotal to carcinogenesis, and TK1, a cell cycle and apoptosis-related kinase, although not a protein kinase. Among the potential protein targets of TunC, these two kinases were selected based on specific criteria: (i) the availability of an empirically derived three-dimensional structure, (ii) high-throughput docking scores, (iii) site map analysis (not included in this manuscript) indicating a cavity large enough to accommodate the bulky TunC ligand without significant strain, and (iv) stability within the TIP3P explicit solvent model used for molecular dynamics (MD) simulations. We postulated that TunC would interact with these targets, yielding a negative ∆Gbind, indicative of spontaneous binding. Computational modelling was employed to evaluate the interaction between TunC and TK1/PKAc, as well as its potential impact on the conformational landscape of these proteins. We hypothesised that TunC binding could modulate the functional properties of these enzymes, potentially altering their roles in carcinogenesis and, indirectly, glycosylation.

PKA is a key regulator of cellular processes, including protein phosphorylation and glycosylation. The cAMP-PKA-CREB signalling pathway plays complex roles in cancer, influencing various cellular processes and the tumour microenvironment [39]. In colorectal cancer, the interplay between cAMP-PKA/EPAC signalling and the TGFβ/SMAD4 pathway regulates stemness and metastatic potential, highlighting its critical role in cancer progression [40]. Additionally, post-translational modifications, such as the crotonylation of PRKACA, enhance PKA activity and drive colorectal cancer development through the PKA-FAK-AKT pathway [41]. This underscores the importance of PKA in promoting tumorigenic signalling cascades. The diagnostic potential of extracellular PKA (ECPKA) in serum further emphasises its relevance in oncology. Elevated ECPKA levels have been linked to gastric and colorectal cancers, presenting a promising biomarker for early detection [42]. Together, these findings demonstrate the intricate involvement of cAMP-PKA-CREB signalling in cancer biology, from regulating the tumour microenvironment to driving metastasis and offering diagnostic value, making it a critical focus for therapeutic development. Glycosylation intersects with PKA pathways, contributing to cancer progression by enhancing tumour cell survival, metastasis, and immune evasion. The intricate relationship between PKA signalling and glycosylation underscores their potential as therapeutic targets [43,44]. Exploring these interconnected pathways offers novel opportunities for developing cancer treatments that disrupt tumour-promoting mechanisms.

TK1 has been identified as one of the top twenty targets obtained from the SwissTarget Prediction program (Table 1). It has been identified as a potential clinical biomarker for the treatment of colorectal cancer, among others, with studies suggesting its upregulation in CRC patients compared to healthy individuals [45]. TK1 was also selected because it is a novel therapeutic target for CRC [34]. TK1 is vital in cellular proliferation and DNA repair [46] and has been implicated in tumour development and progression. Elevated TK1 levels have previously been detected in CRC tissues and are associated with poor prognosis. This elevation suggests that TK1 might contribute to the increased DNA synthesis and replication required for rapid tumour cell proliferation [47]. In addition, it is considered that TK1 may play an indirect role in glycosylation because it is involved in nucleotide sugar synthesis. These sugars, being essential glycosylation precursors, are required to transfer sugar moieties to proteins during glycosylation [48]. Since thymidine kinase activity can influence the availability of nucleotide sugars, this potentially affects glycosylation patterns [49].

The crystal structure of TK1 with dTTP has been elucidated [26]. However, the results are different to those of the published experimental structure of TK1 with regard to the binding site. When comparing the size and structural characteristics of dTTP to Tunicamycin C, it becomes evident that they differ significantly, with Tunicamycin being a much larger, bulkier molecule. Due to its size and hydrophobic tail, Tunicamycin will likely act as a channel blocker. This characteristic may explain why Tunicamycin prefers binding within the subunit interface rather than occupying the dTTP binding site. Regarding the binding mode of Tunicamycin in 5O5E, which represents the human UDP-N-acetylglucosamine-dolichyl-phosphate *N*-acetylglucosamine phosphotransferase (DPAGT1) V264G mutant in complex with Tunicamycin [50], it is likely that the ligand’s flexibility, due to its numerous rotatable bonds, allows it to adapt its orientation to bind to a variety of cavities in proteins. This structural flexibility may contribute to the reported toxicity of Tunicamycin [51,52], as it enables the compound to interact with multiple molecular targets. Such promiscuous binding behaviour is characteristic of compounds with broad activity, but it also poses challenges for therapeutic applications. Consequently, ongoing research aims to reduce Tunicamycin’s toxicity by modifying its structural characteristics. These efforts focus on refining the ligand’s design to maintain its therapeutic potential while minimising off-target effects and adverse interactions with other proteins.

In this paper, we highlight the glycosylation pathways that Tunicamycin C targets. Importantly, as the glycosylation inhibitors were not the top 20 targets for Tunicamycin, they were not selected for further analysis. Nevertheless, we do demonstrate here Tunicamycin C targets involved in glycosylation pathways, such as galactose metabolism, starch and sucrose metabolism, and other types of O-glycan biosynthesis. Tunicamycin C can potentially reduce CRC progression through several mechanisms by targeting hexokinases (HK1, HK2), and hence, glycolytic pathways are disrupted, therefore reducing the energy supply to cancer cells, since cancer cells often rely on aerobic glycolysis (the Warburg effect) for energy [53,54]. Tunicamycin C thus has the potential to inhibit these enzymes and starve cancer cells, slowing their growth and proliferation.

Tunicamycin C might be a potential inhibitor for Beta-1,4-galactosyltransferase (1B4GALT1) and MGAM (Maltase-Glucoamylase) (Appendix A). Beta-1,4-galactosyltransferase is involved in the synthesis of glycoproteins and glycolipids, which play a role in cell–cell and cell–matrix interactions, signalling, and metastasis [55]. MGAM is also involved in carbohydrate digestion and absorption [56]. Therefore, Tunicamycin C will most likely alter glycosylation patterns on the cell surface, potentially reducing cancer cell adhesion, migration, and invasion. ST6 Beta-Galactoside Alpha-2,6-Sialyltransferase 1 (ST6GAL1) and O-GlcNAc transferase (OGT) were also identified as potential targets for Tunicamycin C. ST6GAL1 is involved in sialylation [57], and OGT adds N-acetylglucosamine to proteins, which can disrupt the biosynthesis of O-glycans [58]. This disruption can affect protein stability and signalling pathways, impairing cancer cell survival and proliferation. Altered glycosylation patterns can affect the immune system’s ability to recognise and destroy cancer cells by targeting enzymes involved in glycosylation. Inhibitors can potentially enhance the immune response against cancer cells. Moreover, altered glycosylation patterns can affect the immune system’s ability to recognise and destroy cancer cells. By targeting enzymes involved in glycosylation, inhibitors can potentially enhance the immune response against cancer cells. Additionally, specific glycosylation patterns are associated with inflammation, which can promote cancer progression. Inhibitors altering these patterns may reduce inflammation, slowing cancer progression [59].

The docking results show that the configuration of the docked TK1 exhibited an extreme alignment in terms of conformational orientation with that obtained from the crystal structures from the RCSB PDB. A 2D image of the interaction of the active residues of PKAc with Tunicamycin C in the crystal state and when docked shows that the docked complex exhibited a higher/better hydrogen interaction than the crystal complex, and here, the apo systems exhibited subtle movement, whereas the Tunicamycin C-bound system was slightly more stable than the apo form. A critical examination of this RMSD pattern suggests that the binding of Tunicamycin C could not produce a marked reduction or increase in RMSD even when the binding reduced trajectory fluctuation, which could be suggestive of a favourable binding interaction [60]. High RMSD correlates with instability, while low RMSD indicates stability [61]. Our results suggest that as the binding of Tunicamycin C did not produce a marked change in the RMSD, no firm conclusions can be drawn from this analysis. However, the minor difference in conformational deviation reflected in the plots for TK1 cannot adequately represent stability. However, it is judged as structural instability, which correlates to functional inactivity. Thus, the binding of Tunicamycin C resulted in TK1 inactivity, which could very likely halt cancer cell proliferation.

The plots of the RMSF depict a rigid fluctuation pattern in both TK1 and PKAc, considering the near similar trajectory pattern between the Tunicamycin C-bound systems and the apo. Comparing this to the result obtained for the RMSD plot, the binding of Tunicamycin C induced a structurally rigid system, which indicates inhibition of TK1 activities. The RMSD plot results indicate that the binding of Tunicamycin C to PKAc produced a perturbation that induced a significant 3D structural deviation of PKAc. The substantial divergence in the Tunicamycin C-bound PKAc trajectory compared to its apo form, with the former characterised by higher RMSD, signifies potential functional inactivity. This divergence suggests that Tunicamycin C binding could trigger structural alterations, which might potentially arrest the progression of CRC.

Exploring the RMSF plot trajectory for PKAc and TK1 reveals a notable similarity relative to their apo forms between the Tunicamycin C-bound TK1 and PKAc fluctuation patterns. There is no significant difference in these patterns. Specifically, in the case of PKAc, Tunicamycin C binding appears to induce residue rigidity, indicating negligible variation in RMSF values. The data thus far suggest that Tunicamycin C binding results in a negative structural deviation in both proteins. This disruption could potentially inhibit the activities of both proteins, thereby suppressing cancer cell proliferation.

The cartoon and surface representations reveal that Tunicamycin C aligned well in the binding grooves of both TK1 and PKAc, which might have created the interactions that could have produced dynamics that imply inhibition. The Rg plot demonstrates that the trajectory of Tunicamycin C-bound TK1 and the apo system projects a similar trajectory trend. However, the Tunicamycin C-bound TK1 system appears to be exhibiting a lower trajectory trend towards the end of the MD simulation. The inference drawn from both the plot and the mean Rg value is that the binding of Tunicamycin C could not induce much perturbation in the 3D structure of TK1. This could be due to the mechanism of action of TK1, as the observed deviation in Tunicamycin C-bound TK1 did not reveal much structural difference relative to that of the apo protein. The Rg value of PKAc reveals similar trajectory patterns as that of TK1, wherein the binding of Tunicamycin C to PKAc and the apo form has a similar Rg trajectory that cannot reflect compactness. In most dynamic studies, the binding of ligands usually produces a trajectory pattern whereby the ligand-bound system exhibits lower RMSD and Rg, which are quite distinct from the apo protein; this indicates stability and suggests functional activation [62]. However, in the present study, the ligand-bound systems exhibited a type of trajectory, which trailed that of the apo proteins, indicating a lack of conformational stability rather than reflecting instability, which indicates inactivity. Thus, the binding of Tunicamycin C induces a conformational variation that results in structural changes that inhibit TK1 and PKAc, which halts cancer cell proliferation.

However, in assessing the trajectory of Tunicamycin C-bound PKAc, the fluctuation was slightly more significant than that for Tunicamycin/TK1. This could presumably be due to the reorientation of the larger Tunicamycin C molecule for an appropriate accommodation that would conform to the initially reported fitness/orientation conformation at the pocket of PKAc, which has a smaller 3D structure relative to TK1. In addition, the pharmacophoric region of Tunicamycin C maintained a similar orientation. Moreover, the higher ligand RMSD of the PKAc relative to TK1 suggests a greater ability to inhibit PKAc than TK1. While this drift of TK1 and PKAc at 140 ns of the MD run may reflect the internal dynamics of the systems, this is beyond the scope of this study.

The MSA plot portrays a similar trajectory trend in both Tunicamycin C-bound TK1 and PKAc cases, and shows that the overall dynamics of the polar surface of the ligand remained constant across the two proteins throughout the MD simulation. The MSA we obtained could suggest an adequately established hydrophobic interaction promoting ligand binding. The SASA result reveals that the surface exposure of atoms in TK1-bound Tunicamycin C was lowered greatly relative to that of PKAc across the MD simulation time. The results obtained for ligand RMSD and SASA suggest that Tunicamycin C is a better ligand for PKAc than TK1. SASA indicates and equally reveals hydrophobic interaction and hydrophobicity, which promotes ligand interactions. The plots show that the trajectory of the SASA plots of the binding of Tunicamycin C to TK1 and PKAc maintained a presumably steady type of motion throughout the MD simulation time. Since the simulation plot did not exhibit a wide shift in trajectory, this suggests that at no point did the ligands attempt to move out of the binding pockets. Thus, a steady configurational orientation was obtained for the ligands, as ligand stability was demonstrated within the pocket. The trajectory of this plot indicates that the buried surface-exposed moieties of Tunicamycin C transitioned to the hydrophobic core more in TK1 than in PKAc, which promotes the interaction of Tunicamycin C with these proteins. This trajectory remained constant during the MD simulation. This suggests a steadily sustained ligand binding interaction in the pockets of TK1 rather than PKAc.

The nature and types of interactions established between Tunicamycin C and TK1 and PKAc could provide further insight into the silent structural events responsible for the observed binding stabilisation of Tunicamycin C within the pockets of the two proteins. We noted that the presence of bulk water promoted the formation of higher hydrogen bonding compared to the docked complexes. Additionally, hydrogen bonding enhanced the ligand interactions, molecular recognition, and protein targeting.

Within this analysis, the active site residues that primarily played crucial roles in the binding stabilisation of Tunicamycin C with their corresponding percentage occupancy were represented. For TK1, they are as follows: Gln20, Gln22, Arg38, Arg42, Tyr48, Thr150, Phe157, Arg158, and Glu159. Regarding PKAc, Ser53, Asp182, Lys72, Cys199, Asp166, Gln84, and Phe187 (which makes π-π* interaction) were important for stabilisation. There were a few cases where two arrows showed two different percentage occupancies, which have also been enclosed in a bracket; these are points where two different moieties in the amino acid interacted with one atom in the ligand. Notably, the presence of bulk water in both average PDBs contributed equally to binding stabilisation, thus consolidating the prior assertion regarding the critical role of water in the binding stabilisation of ligands. Collectively the water bridge, van der Waals, and H-bond interactions predominantly established the contacts between the ligands and the active site residues.

From the per-residue energy decomposition analysis, we deduced that Arg38, Arg39, Arg42, and Glu159 in TK1 contributed their respective quota of electrostatic energy to the binding stabilisation of Tunicamycin C. At the same time, for PKAc, the role of Glu69, Lys79, Lys83, and Leu104 also provided a good quantum of electrostatic energy to its binding stabilisation. It is equally important to notice that although more residues in TK1 contributed to the binding stabilisation of Tunicamycin C, the energy output in both proteins was similar. It can be seen from the MM/GBSA result obtained that Tunicamycin C interacted with the active site residues of the two proteins to produce an extremely high ∆*G*_bind_ PKAc, producing favourable binding.

Overall, the PCA results for both PKAc and TK1 proteins indicate that Tunicamycin C treatment leads to notable structural and dynamic alterations in these proteins, including conformational adjustments, increased regional flexibility, and variations in overall compactness, all of which may influence the biological functions of PKAc and TK1. These findings emphasise the significant influence of Tunicamycin C on protein structure and dynamics, providing insights into its potential mechanisms of action and effects on protein function, thus contributing to our understanding of how this compound interferes with cellular processes and protein stability.

## 5. Conclusions

We have adopted an in silico technique to retrieve potential receptor proteins for Tunicamycin C, a promising ligand for ameliorating CRC. Several target proteins with functional roles in cancer-related pathways were identified. The observed structural and interaction events presented Tunicamycin C as a promising inhibitor of both TK1 and PKAc.

While these in silico analyses provide invaluable predictive models for protein functionality, confirmation is required using in vitro and in vivo models.

The observed conformational changes and increased flexibility in PKAc and TK1 proteins suggest that Tunicamycin disrupts their structural integrity, potentially leading to altered protein function. Variations in compactness further indicate that Tunicamycin affects these proteins’ overall folding and shape, which could have significant implications for their biological activities. TK1 plays a known role in CRC and may be important for glycosylation. Glycosylation of proteins, an essential post-translational modification process, affects cancer development and progression via key cellular processes. Although no direct evidence links TK1 and glycosylation in CRC, both are important for cellular functioning and cancer progression.

These findings are consistent with current research on the effects of Tunicamycin on protein structure. Studies have shown that Tunicamycin, through its inhibition of N-linked glycosylation, can induce endoplasmic reticulum stress and affect protein folding and stability. The deductions made from this study serve as a framework that can provide an informed decision in an experimental approach for in vitro and in vivo investigations of TK1 and PKAc as potential receptor targets for Tunicamycin C in the control of CRC development and progression. Understanding the structural events underlying Tunicamycin’s C interactions with these targets would provide valuable insights for designing effective drugs in CRC treatment. Overall, exploring specific glycosylation targets represents a paradigm shift in cancer therapy and holds promise for improving treatment outcomes in CRC.

## Figures and Tables

**Figure 1 cimb-47-00339-f001:**
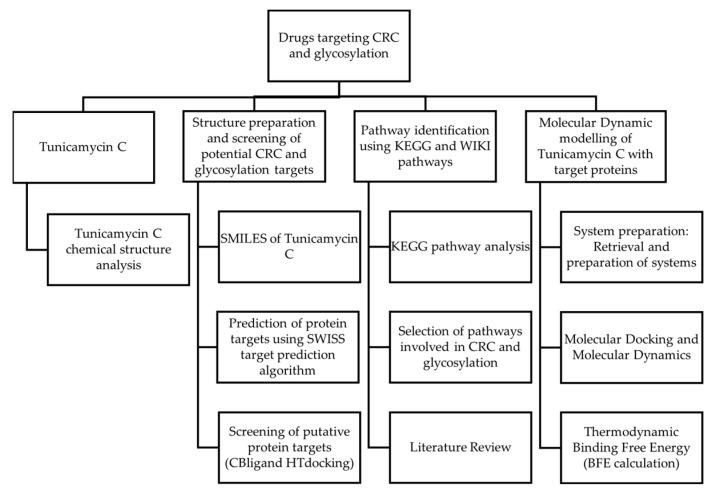
Research design.

**Figure 2 cimb-47-00339-f002:**
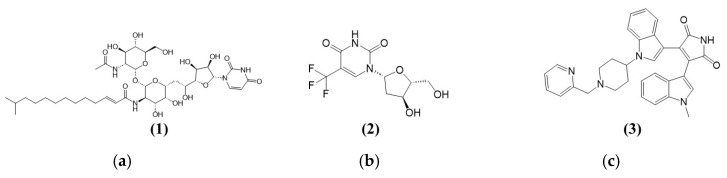
Chemical structures of (**a**) (**1**) Tunicamycin C (**1**), (**b**) (**2**) positive control for TK1—Trifluorothymidine, and (**c**) (**3**) positive control for PKAc—Bisindolylmaleimide I (GF109203X).

**Figure 3 cimb-47-00339-f003:**
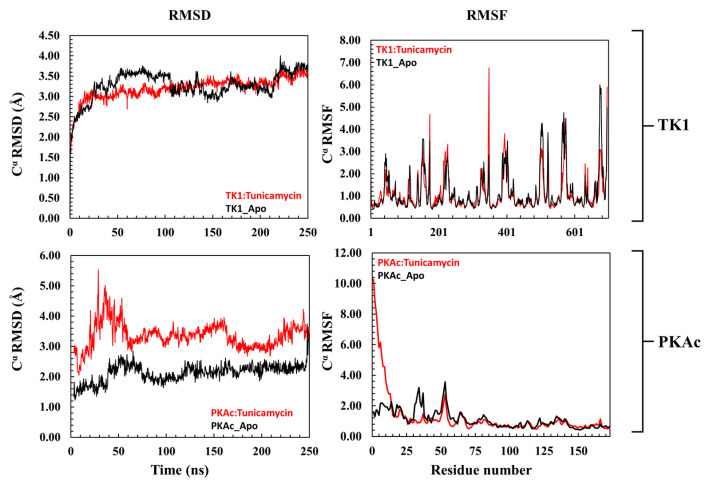
A representation of the Cα-RMSD (root mean square deviation) and Cα-RMSF (root mean square fluctuation) of TK1 (**top panel**) and PKAc (**bottom panel**) with their corresponding apo. The upper plots are a representation of the trajectory pattern for TK1, while the lower plots are for PKAc.

**Figure 4 cimb-47-00339-f004:**
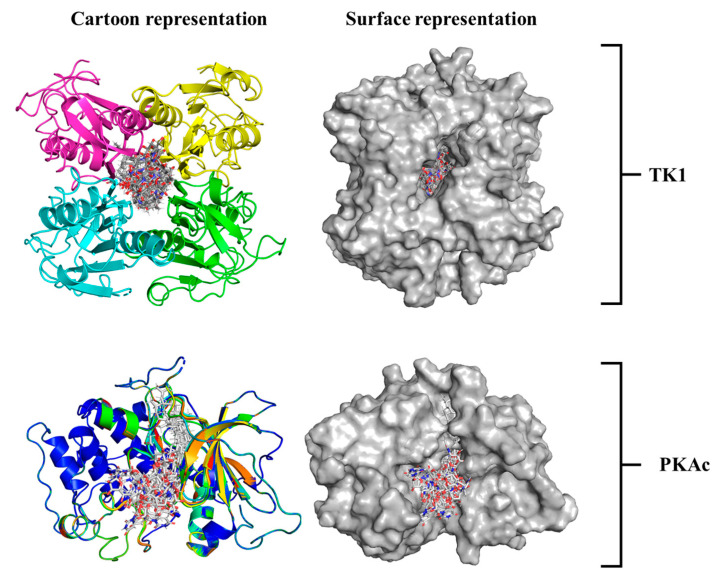
An illustrative presentation of Tunicamycin C TK1(**top panel**) and PKAc (**bottom panel**) docking complexes showing the best binding pose conformation for optimal interaction.

**Figure 5 cimb-47-00339-f005:**
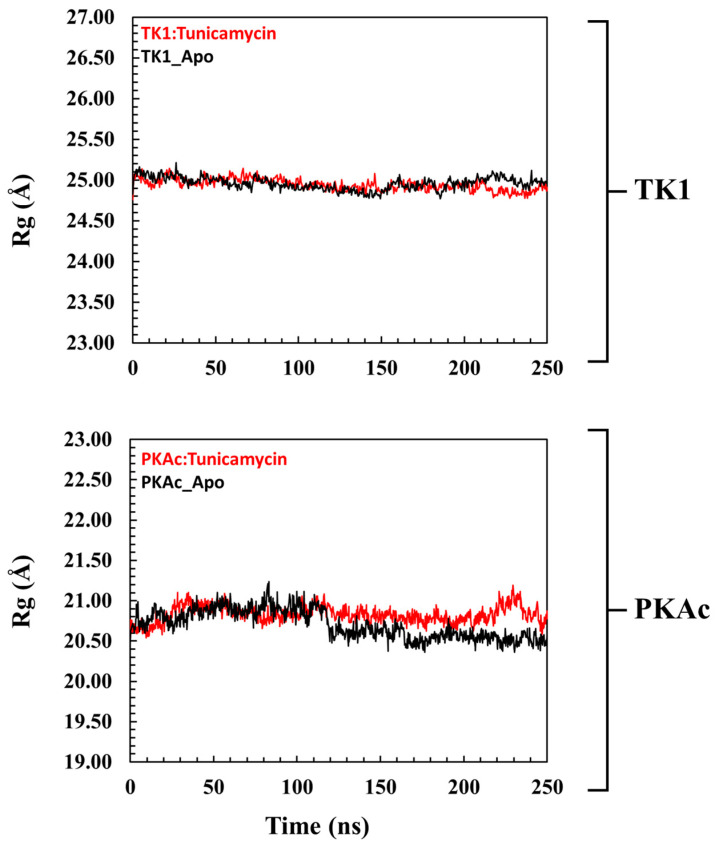
A pattern representation observed in the Rg plots of the global structure of (**top panel**) TK1 and (**bottom panel**) PKAc with their corresponding apo systems.

**Figure 6 cimb-47-00339-f006:**
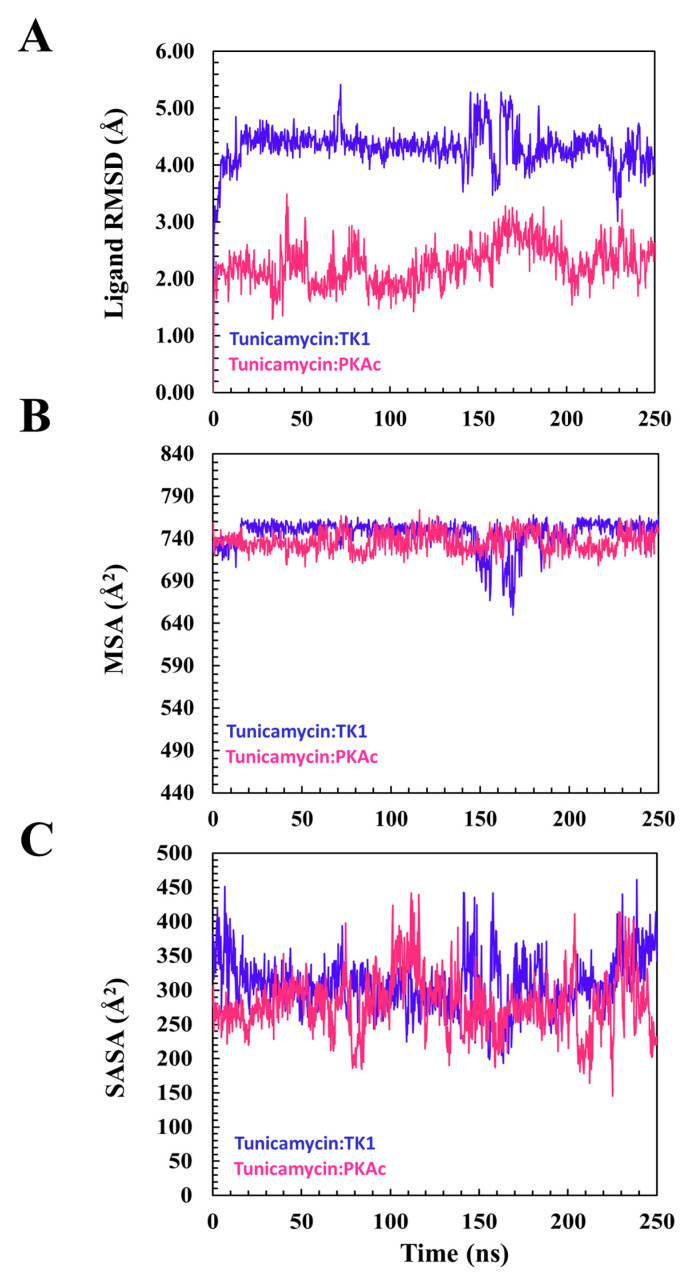
A descriptive indication of Tunicamycin C performance during the MD timeline. This gives an overview of the ligand’s interaction and stabilisation pattern. (**A**) The root square mean deviation for TK1 and PKAc during the modelling of Tunicamycin binding. (**B**) The molecular surface area of the modelled interaction. (**C**) The accessible solvent surface area for the modelled interaction.

**Figure 7 cimb-47-00339-f007:**
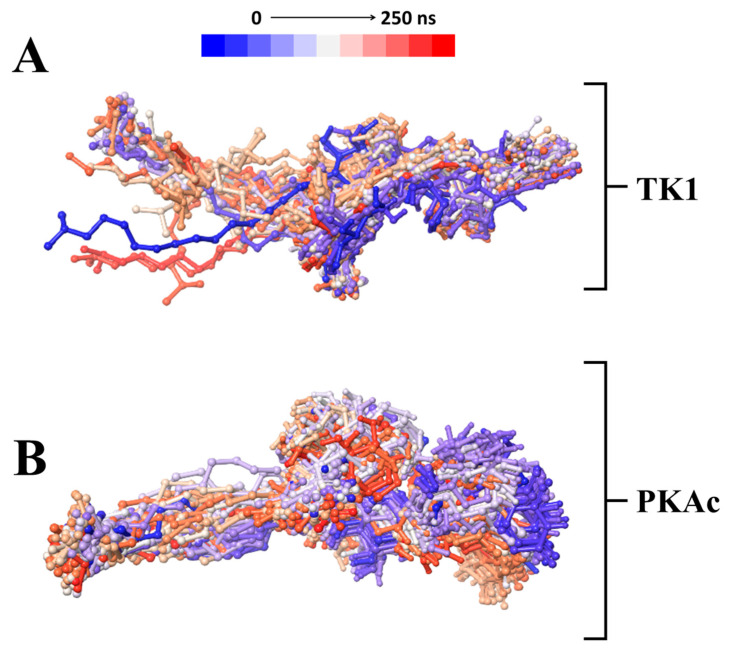
A time-based structural presentation of ligand conformation and orientation during the MD simulation of TK1 (**A**) and PKAc (**B**).

**Figure 8 cimb-47-00339-f008:**
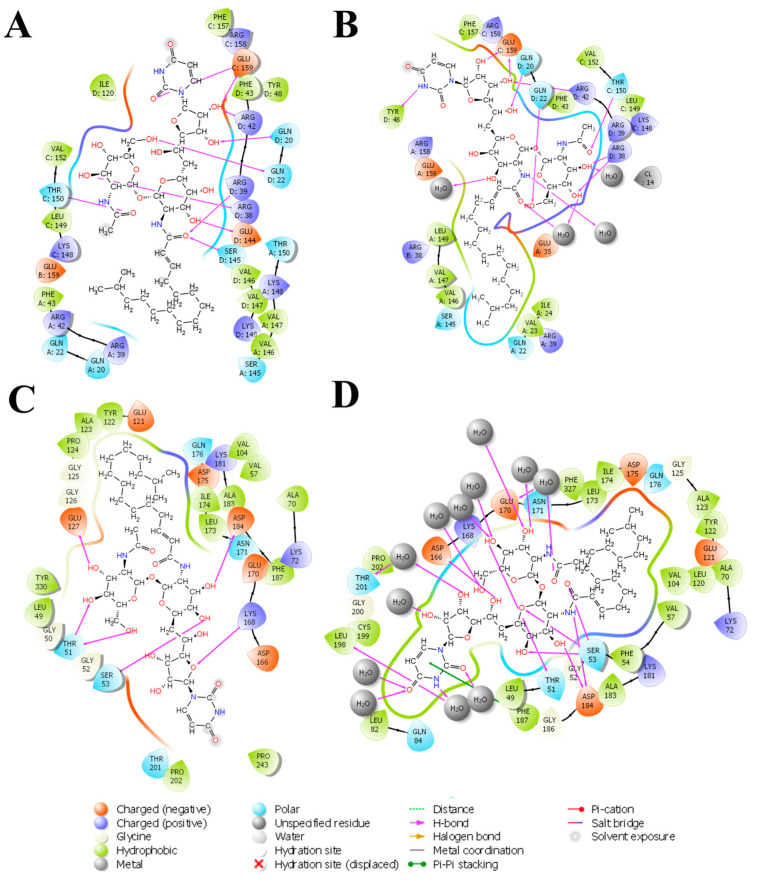
The 2D interaction representation of Tunicamycin, TK1, and PKAc complexes shows the difference between the docking and MD complex environments. (**A**,**C**) The docking complex environment; (**B**,**D**) the explicit solvent environment, which provides an idea of the role of bulk water, depicting a typical physiological environment for better binding interaction stabilisation.

**Figure 9 cimb-47-00339-f009:**
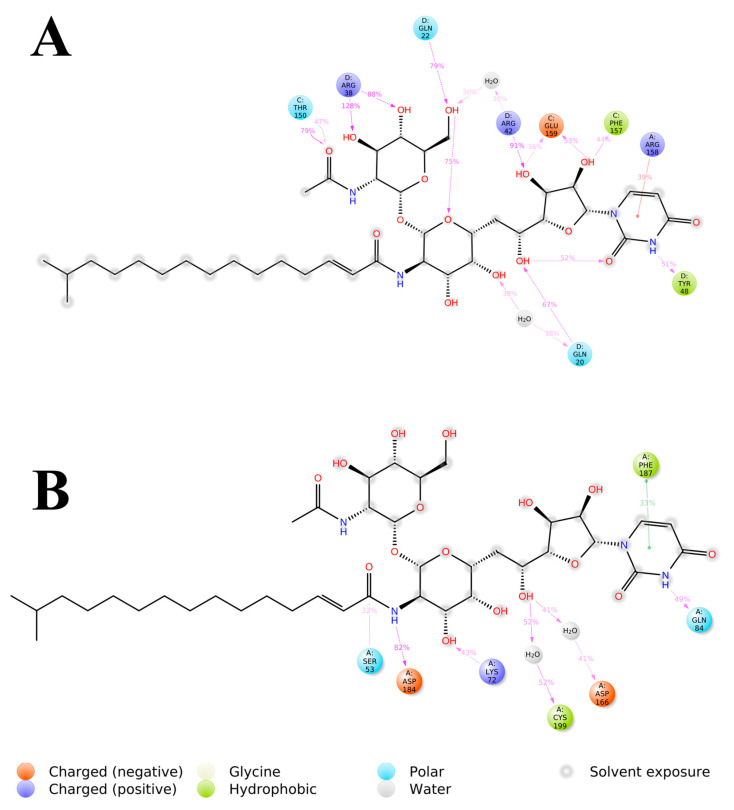
The 2D representation of Tunicamycin interactions with (**A**) TK1 and (**B**) PKAc. The strongest, most dominant interactions are represented in the average PDB model. The most consistent interactions between the side chains and Tunicamycin C, are shown.

**Figure 10 cimb-47-00339-f010:**
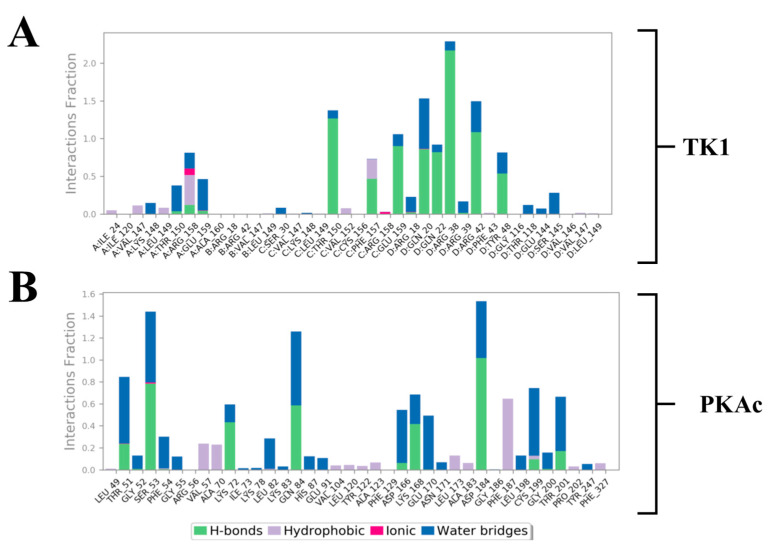
Stacked bar charts of side-chain interactions and interaction types between Tunicamycin and (**A**) TK1 and (**B**) PKAc. The H-bond is categorised into backbone acceptor, backbone donor, side-chain acceptor, and side-chain donor. Hydrophobic interactions are categorised into π-cation, π-π*, and non-specific interactions.

**Figure 11 cimb-47-00339-f011:**
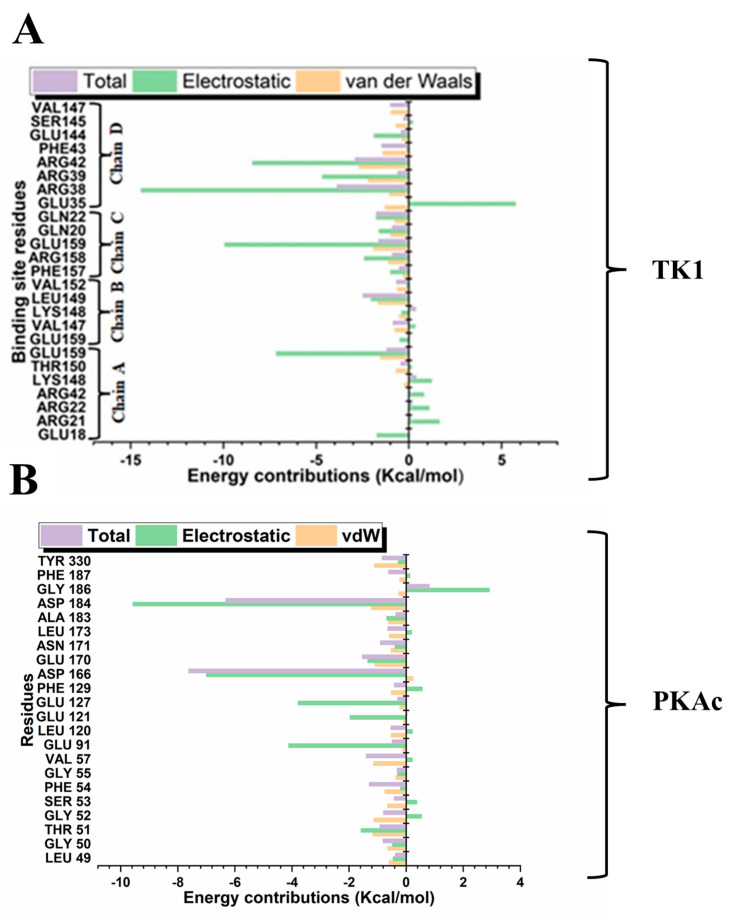
Per-residue energy decomposition for the interaction in Tunicamycin and (**A**) TK1 and (**B**) PKAc systems.

**Figure 12 cimb-47-00339-f012:**
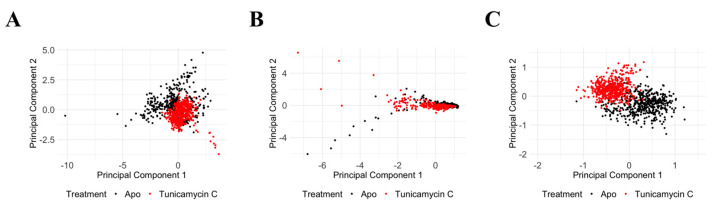
PCA of PKAc protein under apo and Tunicamycin-treated conditions. (**A**) PCA of C_α_ RMSD data between apo (black) and Tunicamycin-treated (red) proteins. PC1 accounts for 74.49% of the variance, while PC2 accounts for 25.51%. (**B**) PCA of C_α_ RMSF data indicating some overlap, but with distinct shifts between apo (black) and Tunicamycin-induced (red) forms. PC1 explains 89.96% of the variance, and PC2 explains 10.04%. (**C**) PCA of radius of gyration (RoG) data highlighting distinct clustering for apo (black) and Tunicamycin-treated (red) forms. PC1 accounts for 98.66% of the variance, with PC2 accounting for 1.34%.

**Figure 13 cimb-47-00339-f013:**
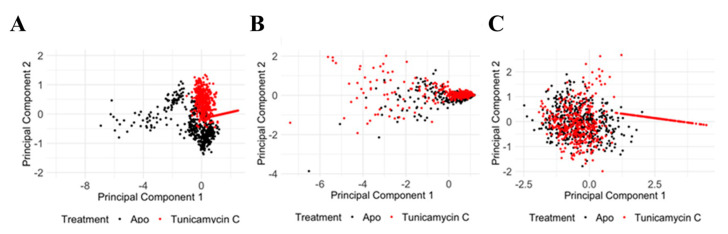
PCA of TK1 protein under apo and Tunicamycin C-induced conditions. (**A**) PCA of C_α_ RMSD data showing a clear separation between apo (black) and Tunicamycin-treated (red) proteins. PC1 accounts for 83.86% of the variance, while PC2 accounts for 16.14%. (**B**) PCA of C_α_ RMSF data between apo (black) and Tunicamycin C-treated (red) proteins. PC1 explains 90.75% of the variance, and PC2 explains 9.25%. (**C**) PCA of RoG data highlighting distinct clustering for apo (black) and Tunicamycin-treated (red) forms. PC1 accounts for 80.70% of the variance, while PC2 accounts for 19.30%.

**Table 1 cimb-47-00339-t001:** Potential target proteins for Tunicamycin C as predicted from the SwissTarget Prediction platform.

Target	Common Name	Uniprot ID	Known Actives	HTDocking Score
**Tunicamycin C targets**				
Cytidine deaminase	CDA	P32320	0/2	6.8
Protein farnesyltransferase	FNTA FNTB	P49354 P49356	2/0	7.7
Apoptosis regulator Bcl-X	BCL2L1	Q07817	1/0	8.2
Apoptosis regulator Bcl-2	BCL2	P10415	1/0	8.2
Thymidine kinase, cytosolic	TK1	P04183	0/13	7.4
Protein Kinase A catalytic subunit	PKAc	P17612	0/1	6.8
**Trifluorothymidine target validation**				
Thymidine kinase, cytosolic	TK1	P04183	0/18	6.5
**Bisindolylmaleimide I (GF109203X)**				
Protein kinase C beta	PKCb	P05771	233/135	

**Table 2 cimb-47-00339-t002:** KEGG pathways analysis of Tunicamycin C protein targets involved in CRC.

#Term ID	Term Description	Gene Count	False Discovery Rate	Matching Proteins in Your Network (Labels)
hsa04014	Ras signalling pathway	11	7.94 × 10^−7^	FLT3, FGF2, PKAc, PLA2G5, BCL2L1, PLA2G2A, PLA2G10, PRKCA, FGF1, **PKCb**, AKT3
hsa05205	Proteoglycans in cancer	10	1.75 × 10^−6^	MMP2, FGF2, PDCD4, PKAc, ROCK2, MMP9, HPSE, PKAc, **PKCb**, AKT3
hsa05200	Pathways in cancer	13	2.49 × 10^−5^	MMP2, FLT3, FGF2, PKAc, ROCK2, HSP90AA1, MMP9, BCL2L1, BCL2, PKAc, FGF1, **PKCb**, AKT3
hsa01521	EGFR tyrosine kinase inhibitor resistance	6	5.42 × 10^−5^	FGF2, BCL2L1, BCL2, PKAc, **PKCb**, AKT3
hsa05206	MicroRNAs in cancer	6	0.0014	PDCD4, PKCE, MMP9, BCL2, PKAc, **PKCB**
hsa04151	PI3K-Akt signalling pathway	8	0.0027	FLT3, FGF2, HSP90AA1, BCL2L1, BCL2, PKAc, FGF1, AKT3
hsa05230	Central carbon metabolism in cancer	4	0.0033	FLT3, HK2, HK1, AKT3
hsa04010	MAPK signalling pathway	7	0.0037	FLT3, FGF2, PKAc, PKAc, FGF1, PRKCB, AKT3
hsa04370	VEGF signalling pathway	3	0.0146	**PKAc**, **PKCB**, AKT3
hsa04310	Wnt signalling pathway	4	0.0267	PRKACA, ROCK2, **PKAc**, **PKCB**
hsa04630	JAK-STAT signalling pathway	4	0.0284	PTPN2, BCL2L1, BCL2, AKT3
**KEGG pathways implicated in glycosylation**				
hsa00052	Galactose metabolism	6	7.94 × 10^−7^	SI, HK2, GAA, B4GALT1, MGAM, HK1
hsa00500	Starch and sucrose metabolism	7	4.60 × 10^−8^	PYGM, SI, HK2, GAA, MGAM, AMY2A, HK1
hsa00514	Other types of O-glycan biosynthesis	3	0.0082	ST6GAL1, OGT, B4GALT1

**Table 3 cimb-47-00339-t003:** Estimated binding energy predictions of Tunicamycin complexes.

Protein Complex	Energy Components (kcal/mol)				
	∆*E*_vdw_	∆*E*_ele_	∆*G*_gas_	∆*G*_sol_	∆*G*_bind_
TK1_Tunicamycin C	−69.13 ± 4.25	−70.08 ± 15.05	−139.20 ± 14.67	78.99 ± 9.96	−60.21 ± 9.46
PKAc_Tunicamycin C	−37.05 ± 8.81	−110.32 ± 17.46	−147.37 ± 16.75	106.78 ± 9.54	−40.59 ± 9.94

## Data Availability

The original contributions presented in this study are included in the article/Appendix A. Please direct further inquiries to the corresponding author.

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
