# Peer review of "Computational Modelling of Tunicamycin C Interaction with Potential Protein Targets: Perspectives from Inverse Docking with Molecular Dynamic Simulation"

_cimb, 2025, doi:10.3390/cimb47050339_

Round 1

Reviewer 1 Report (New Reviewer)

Comments and Suggestions for Authors
  1. Improve quality (readability) of figure 1
  2. line 202 - space after reference, check throughout the manuscript unnecessary spaces 
  3. Improve quality of figure 2
  4. explain all abbreviations under the table 1
  5. what is the gap before table 2? Also use full word for the table or figure title
  6. Line 429 - space is missing, check whole manuscript 
  7. add abbreviations under figure 3
  8. figure 11 - different quality of 2 graphs, improve 
  9. figure 12 - write title in one sentence, the rest of the text put in manuscript 
  10. please do not refer to figures and tables in the discussion section (use term our results of…, not term as indicated in figure etc)
  11. please add limitation section 

Author Response

Thank you for your review. Please find our reply in the attachment

Reviewer 2 Report (New Reviewer)

Comments and Suggestions for Authors

1- Please improve the resolution of figure 1. 

2- There are several typos and grammatical errors within the manuscript. Please revise the manuscript text carefully. 

3- Please cite all protocols and tools used in the M&M section

4- The output of DSSP in GROMACS should be added to the content. The respected authors should show how the interaction between ligand and receptor active site pocket affect the structural features of protein. 

5- Figure 8: Please confirm the respected authors has valid license to use Schrodinger computational suite. If you used a cracked version, this case can cause problem after publishing this paper. 

6- What is the novelty of this paper? 

7- Please supplement figure 10. Excessive number of figures were used within the content of this paper. 

8- Please merge figures 12 & 13

Author Response

Thank you for your review. Please find our reply in the attachment

This manuscript is a resubmission of an earlier submission. The following is a list of the peer review reports and author responses from that submission.

Round 1

Reviewer 1 Report

Comments and Suggestions for Authors

1. line 4: Could the author clarify the meaning of the term 'cum' used in the title?

2. line 48: It would be helpful to include a few references for targeted therapy in cancer to support the statement.

3. line 145: Could the author clarify the meaning of 'zero bond orders to metals'?

4. line 256: Could the author explain the rationale behind the expressions labeled as equations (3) and (4), as they do not appear to be traditional equations?
5. line 300: Could the authors elaborate on how the binding site of Tunicamycin C was determined for TK1 and PrKC1? 

6. line 327: According to Figure 2, the binding site of Tunicamycin C on TK1 appears to be in the middle pore of the TK1 tetramer. However, based on the TK1 crystal structure (PDB: 1XBT), there is a dTTP molecule bound to TK1, but not in the middle pore. Additionally, the docked binding site seems quite different compared to known Tunicamycin binding sites in crystal structures such as PDB 5O5E and 6BW5. Could the authors comment on these differences and elaborate how they determine the binding site on TK1 and PrKC1 for tunicamycin C?

Author Response

  1. line 4: Could the author clarify the meaning of the term 'cum' used in the title?

This has been addressed. We thank the reviewer.

  1. line 48: It would be helpful to include a few references for targeted therapy in cancer to support the statement.

This has been addressed. We thank the reviewer.

  1. line 145: Could the author clarify the meaning of 'zero bond orders to metals'?

"zero bond orders to metals" refers to the absence of significant covalent bonding interactions between a metal atom and other atoms. This occurs when interactions are primarily ionic, involve weak or transient coordination, or when computational methods fail to capture metal-ligand covalent bonds due to parameterisation limitations. Delocalised electronic effects can also obscure bonding. Understanding zero bond orders is essential for interpreting metal-ligand complexes and catalytic systems in computational studies.

We have added a line to further clarify this in the revised manuscript.

  1. line 256: Could the author explain the rationale behind the expressions labeled as equations (3) and (4), as they do not appear to be traditional equations?

This has been corrected. Thank you very much.

  1. line 300: Could the authors elaborate on how the binding site of Tunicamycin C was determined for TK1 and PrKC1? 

After identifying several potential molecular targets for Tunicamycin using online HTDocking tools, we conducted further analysis on each target with well-defined coordinates available in the PDB database. This was performed using the SiteMap module implemented in Maestro v13. We evaluated each protein target for an active site cavity large enough to accommodate the Tunicamycin ligand without causing significant perturbation or strain. Based on this analysis, we decided to use the oligomeric structure of TK1 rather than disaggregating the multimer. For PKAc, the catalytic subunit appeared to be the most likely binding site. Nonetheless, during the more rigorous induced fit docking process, which was performed to obtain a more accurate representation of docking scores and poses for subsequent MD simulations, we did not bias the algorithm toward any specific binding site. Instead, the entire protein was used as the search space for docking studies.

  1. line 327: According to Figure 2, the binding site of Tunicamycin C on TK1 appears to be in the middle pore of the TK1 tetramer. However, based on the TK1 crystal structure (PDB: 1XBT), there is a dTTP molecule bound to TK1, but not in the middle pore. Additionally, the docked binding site seems quite different compared to known Tunicamycin binding sites in crystal structures such as PDB 5O5E and 6BW5. Could the authors comment on these differences and elaborate how they determine the binding site on TK1 and PrKC1 for tunicamycin C?

We thank the reviewer for this comment.

When comparing the size and structural characteristics of dTTP to Tunicamycin C, it becomes evident that they differ significantly, with Tunicamycin being a much larger, bulkier molecule. Due to its size and hydrophobic tail, Tunicamycin is likely to act as a channel blocker. This characteristic may explain why Tunicamycin prefers binding within the subunit interface rather than occupying the dTTP binding site.

Regarding the binding mode of Tunicamycin in 5O5E, which represents the human UDP-N-acetylglucosamine-dolichyl-phosphate N-acetylglucosamine phosphotransferase (DPAGT1) V264G mutant in complex with Tunicamycin, it is likely that the ligand's flexibility, due to its numerous rotatable bonds, allows it to adapt its orientation to bind to a variety of cavities in proteins. This structural flexibility may contribute to the reported toxicity of Tunicamycin, as it enables the compound to interact with multiple molecular targets. Such promiscuous binding behaviour is a characteristic of compounds with broad activity, but it also poses challenges for therapeutic applications. Consequently, ongoing research aims to reduce Tunicamycin's toxicity by modifying its structural characteristics. These efforts focus on refining the ligand's design to maintain its therapeutic potential while minimising off-target effects and adverse interactions with other proteins.

Reviewer 2 Report

Comments and Suggestions for Authors

Using in-silico approaches, the authors explore the potential binding targets of tunicamycin-c, a small molecule with known anticancer properties. This manuscript focuses on potential binding targets of tunicamycin-c that are involved in pathways relevant to colorectal cancer (CRC). The authors start with docking their compound to multiple protein targets using a high throughput pipeline implemented in a public web server.proteins with the highest docking score were used to query public pathway databases to identify potential targets relevant to CRC. The authors finally selected two proteins and studied the effect of tunicamycin binding in these targets using molecular dynamics simulations.

I do think, the authors’ approach merits some novelty in the way they identified the two protein targets, but I find this manuscript lacking sufficient details in the results and necessary controls. Here are my major critics.

1. There are very little details on the high throughput results obtained from HTDocking or Swisstarget. The authors present the final protein targets and their pathways in a table form, but a lot of intermediate filtering steps led to those results and those details should be there as part of the results. To improve the manuscript, I suggest to include how many proteins in total were considered by HTDocking for docking tunicamycin and what percent were retained with a docking score of 8. What were the top scoring targets? How does the statistical distribution of docking score look like? These are just some of the suggestions. More information, the better.

2. its not clear what kind of pathway querry was carried out using KEGG and Wikipathways. Was it over-representation analysis (ORA), where all proteins are given equal importance regardless of the docking score? Did the authors also try a GSEA type of approach with proteins ranked according to their docking scores? Any differences in outcome between the two approaches?

3. Table 1 should be accompanied by the docking scores to better assess how the two proteins were selected out of 14.

4. While I leave it to the authors, a graphical schematic representation of the pipeline to identify the top protein targets would increase the impact and clarity of the manuscript. These details are currently only in the Methods text.

5. The authors present a purely in-silico approach for selecting the top protein targets. Without any control, it’s hard to assess the confidence of their approach. I would strongly suggest to perform the same procedure on a few known compounds and show that the top proteins/pathways are enriched by their known binding targets/functional effects. Without this, it’s somewhat shooting in the dark, especially since their results are not backed up by any experimental validation.

6. The MD methods section seems to have some confusions. There are two subsections: “MM/GBSA free binding energy calculation” (should be binding free energy) and “Thermodynamic binding free energy (BFE) calculation”. These two sections seem to indicate the same thing. The first section describes explicit solvent simulations and not free energy calculations. The second section describes MM/GBSA, but it’s not clear which trajectories (Desmond or AMBER) were used here. The authors say, ‘last 250 ns’ were used, but then AMBER simulations were only 25 ns as stated. If Desmond trajectories were used for free energy calculations, what is use of the AMBER simulations? Please clarify. Also, please specify the total length of the Desmond production simulations. Finally, minor but not so minor, eq 3,4 has all ‘+’ signs. One of them must be a ‘=‘.

7. In figure 1, the RMSD profiles didn’t seem to have converted, since at the end of 250 ns, they were still showing increasing trend.

8. Lines 295-318: Binding of tunicamycin in TK1 caused minimal structural deviation from apo. This could mean that the compound is acting as a strong antagonist of TK1 preventing the binding of other signaling molecules, in other words, inhibiting TK1. On the other hand, the structural instability and binding pose diversity in PrKC1 may indicate that the compound is not energetically stable in the binding pocket and not very compatible with PrKC1 binding in the first place. Hence, I don’t completely agree with the authors’ interpretation that tunicamycin inhibits PrKC1.

9. Lines 384-389: I don’t necessarily agree with what the authors are claiming from fig 5. To me, the TK1 bound poses seem less variable than those in PrKC1. This is also supported by the ligand RMSD plots in fig. 4a.

10. Line 394: ‘conformation during MD …’ - which conformation? Since the ligand conformation is varying during MD.

11. Line 397: ‘implicit physiological condition’ - not clear.

12. The average Pdb is an average of multiple structurally aligned frames from MD. It’s essentially an unphysical structure. It doesn’t make sense to analyze ligand interactions using this pdb, since many interactions will be potentially missed due to unphysical orientation of individual sidechains. Why can’t the authors analyze the frequency of the individual protein-ligand contacts from MD and present the ones with high frequency. That’s a more valid analysis in my opinion. Programs like getcontacts makes it easy to do this.

 13. Authors only present the result of the analysis in Figs 8 and 9 without really explaining what these results mean. E.g. Are there any functional significance of the residues showing the strongest interaction with tunicamycin? Without any synthesis presented, these figures should be omitted.

14. Lines 470-473 completely contradicts what was said in lines 302-305 earlier in the manuscript. Also, Fig. 10a,b are labeled as RMSD and RMSF PCA, but these plots are identical!!

To summarize, this study started as interesting, but slowly lost its appeal over too many analysis being presented without an effective synthesis.

Comments on the Quality of English Language

In many places, conciseness of English and thorough revision are advised.

Author Response

  1. There are very little details on the high throughput results obtained from HTDocking or Swisstarget. The authors present the final protein targets and their pathways in a table form, but a lot of intermediate filtering steps led to those results and those details should be there as part of the results. To improve the manuscript, I suggest to include how many proteins in total were considered by HTDocking for docking tunicamycin and what percent were retained with a docking score of 8. What were the top scoring targets? How does the statistical distribution of docking score look like? These are just some of the suggestions. More information, the better.

We used the online HTDocking algorithm for the initial identification of potential molecular targets of the ligands. A selection criterion was used to screen the resulting targets. This selection criteria (enunciated in the first paragraph of the discussion) enabled us to focus our study on TK1 and PKAc, which was misspelt in the original version of the manuscript in several instances. This was also because of some inconsistencies in the naming and hyperlinking of these online databases. Hence, we have used the more conventional PKAc to refer to the catalytic subunit of Protein Kinase A.

  1. its not clear what kind of pathway querry was carried out using KEGG and Wikipathways. Was it over-representation analysis (ORA), where all proteins are given equal importance regardless of the docking score? Did the authors also try a GSEA type of approach with proteins ranked according to their docking scores? Any differences in outcome between the two approaches?

This query has been answered in the previous question. However, for this revision, we repeated the HT docking using re-modelled Tunicamycin C that took into consideration the spatial organisation of the functional groups as well as generating conformers of the ligands using the LigPrep module implemented in Schrodinger Maestro version 13. This ensured that ligands were better represented to reflect the cytosolic condition based on the net charge of the ligands at pH 7.0±0.2.

  1. Table 1 should be accompanied by the docking scores to assess better how the two proteins were selected out of 14.

We have included the docking scores in Table 1, which were seen as one of the selection criteria. We used a more rigorous induced fit docking to probe the selected protein targets in subsequent analysis.

  1. While I leave it to the authors, a graphical schematic representation of the pipeline to identify the top protein targets would increase the impact and clarity of the manuscript. These details are currently only in the Methods text.

In the revised version, we have included a flow chart diagram to illustrate this.

  1. The authors present a purely in-silico approach for selecting the top protein targets. Without any control, it’s hard to assess the confidence of their approach. I would strongly suggest to perform the same procedure on a few known compounds and show that the top proteins/pathways are enriched by their known binding targets/functional effects. Without this, it’s somewhat shooting in the dark, especially since their results are not backed up by any experimental validation.

In the revised manuscript, we have included two control ligands as depicted in Figure 1.

  1. The MD methods section seems to have some confusions. There are two subsections: “MM/GBSA free binding energy calculation” (should be binding free energy) and “Thermodynamic binding free energy (BFE) calculation”. These two sections seem to indicate the same thing. The first section describes explicit solvent simulations and not free energy calculations. The second section describes MM/GBSA, but it’s not clear which trajectories (Desmond or AMBER) were used here. The authors say, ‘last 250 ns’ were used, but then AMBER simulations were only 25 ns as stated. If Desmond trajectories were used for free energy calculations, what is use of the AMBER simulations? Please clarify. Also, please specify the total length of the Desmond production simulations. Finally, minor but not so minor, eq 3,4 has all ‘+’ signs. One of them must be a ‘=‘.

The Desmond MD simulation engine was used to analyse the conformational landscape of the proteins and the proteins in complex with TunC ligand. The Desmond Molecular Dynamics simulation engine is a powerful tool for protein simulations, offering high accuracy and speed in GPU platforms. Its versatility supports diverse applications, including protein-ligand interactions and conformational analysis. However, the shortfall of the Desmond simulation is that its trajectories cannot be easily transposed to the format that could be used for MMPBSA binding free energy calculations. Hence for this purpose, we attempted to use the Amber simulation engine for 25 ns, which is enough to estimate the relative FEB. In the amber system, we have used precisely a similar ff and solvent systems, which is OPLS and TIP3P, which we used in the Desmond simulation.

  1. In figure 1, the RMSD profiles didn’t seem to have converted, since at the end of 250 ns, they were still showing increasing trend.

The PE vs Time plot (not shown in the manuscript), as well as the simulation quality analysis (implemented in the Maetsro) both, indicated that all four systems attained dynamic equilibrium, and hence, they all converged. Nonetheless, some large and complex proteins, such as the polymeric TK1, will show such characteristics.

  1. Lines 295-318: Binding of tunicamycin in TK1 caused minimal structural deviation from apo. This could mean that the compound is acting as a strong antagonist of TK1, preventing the binding of other signaling molecules and, in other words, inhibiting TK1. On the other hand, the structural instability and binding pose diversity in PrKC1 may indicate that the compound is not energetically stable in the binding pocket and not very compatible with PrKC1 binding in the first place. Hence, I don’t completely agree with the authors’ interpretation that tunicamycin inhibits PrKC1.

While we agree with the reviewer in this regard, we must also consider that an enzyme could be inhibited in several modes, especially if the conformational landscape of the enzyme could be altered upon binding of an inhibitor. This is because inhibitor-induced alterations in the conformational landscape of an enzyme can block active site access, stabilise inactive conformations, or propagate allosteric changes, impairing catalytic efficiency. These changes disrupt dynamic flexibility, essential for enzymatic function, leading to inhibition or inactivation. Such shifts highlight the therapeutic potential of a ligand such as Tunicamycin C in the context of this study, which we validated through molecular dynamics simulations and structural analyses.

  1. Lines 384-389: I don’t necessarily agree with what the authors are claiming from fig 5. To me, the TK1 bound poses seem less variable than those in PrKC1. This is also supported by the ligand RMSD plots in fig. 4a.

In the revised version using the more appropriate model of PKAc, we have shown that Tunicamycin binds to both proteins comparably. Nonetheless, the hydrocarbon moiety of tunicamycin bound to TK1 appear to be more solvent-exposed and not restricted compared to how Tunicmycin binds to PKAc.

  1. Line 394: ‘conformation during MD …’ - which conformation? Since the ligand conformation is varying during MD.

  1. Line 397: ‘implicit physiological condition’ - not clear.

This has been deleted. It was an incomplete or mis-represented sentence.

  1. The average Pdb is an average of multiple structurally aligned frames from MD. It’s essentially an unphysical structure. It doesn’t make sense to analyze ligand interactions using this pdb, since many interactions will be potentially missed due to unphysical orientation of individual sidechains. Why can’t the authors analyze the frequency of the individual protein-ligand contacts from MD and present the ones with high frequency. That’s a more valid analysis in my opinion. Programs like getcontacts makes it easy to do this.

We thank the reviewer for this observation. It has been corrected.

  1. Authors only present the result of the analysis in Figs 8 and 9 without really explaining what these results mean. E.g. Are there any functional significance of the residues showing the strongest interaction with tunicamycin? Without any synthesis presented, these figures should be omitted.

This figure presents two stacked bar charts illustrating the interactions stabilizing Tunicamycin C (TunC) with (i) TK1 and (ii) PKAc. Each bar represents the fraction of simulation time a specific residue interacts with TunC, categorized by interaction type. Green bars indicate hydrogen bonds, which are critical for binding specificity and stability, with significant contributions observed in both TK1 and PKAc. Blue bars represent water bridges, reflecting water-mediated interactions that enhance flexibility and are notably prevalent in PKAc. Red bars depict ionic interactions, though these are relatively sparse, suggesting fewer direct electrostatic interactions. Grey bars highlight hydrophobic interactions, underscoring non-polar stabilization, with some residues showing significant contributions to the binding. Overall, the interaction patterns reveal diverse mechanisms stabilizing the TunC complexes with TK1 and PKAc, reflecting their distinct binding environments and functional roles.

For Fig 9: The first figure shows the per-residue energy decomposition for TK1 (panel A) and PKAc (panel B), highlighting the contributions of specific residues to the stabilization of Tunicamycin. Energy contributions are divided into total energy (purple), electrostatic energy (green), and van der Waals (vdW) energy (orange). In both TK1 and PKAc, several residues exhibit significant contributions, with notable differences in the relative importance of electrostatic and vdW interactions for each protein. For TK1 (panel A), residues such as ARG42, ARG39, and GLN220 contribute strongly through electrostatic interactions, while others like VAL147 and PHE157 have significant vdW contributions. In PKAc (panel B), residues such as TYR330 and ASP166 show prominent electrostatic contributions, while hydrophobic residues like PHE129 and LEU173 stabilize through vdW interactions. When related to the second figure, which categorizes interactions by type (H-bonds, hydrophobic, ionic, and water bridges), it becomes clear that residues contributing strong energy in the first figure are also involved in specific stabilizing interactions. For example, ARG42 and ASP166 show prominent ionic interactions, while hydrophobic residues contribute both vdW energy and hydrophobic contacts. This alignment reinforces the critical role of these residues in stabilizing Tunicamycin binding. Together, Figs 8 and 9 provide a comprehensive understanding of the binding dynamics and energy contributions for both TK1 and PKAc.

  1. Lines 470-473 completely contradicts what was said in lines 302-305 earlier in the manuscript. Also, Fig. 10a,b are labeled as RMSD and RMSF PCA, but these plots are identical!!

This has been corrected.

To summarize, this study started as interesting, but slowly lost its appeal over too many analysis being presented without an effective synthesis.